# Ibrutinib Plus RCHOP versus RCHOP Only in Young Patients with Activated B-Cell-like Diffuse Large B-Cell Lymphoma (ABC-DLBCL): A Cost-Effectiveness Analysis

**Hayeong Rho [1,*] , Irene Joo-Hyun Jeong [1] and Anca Prica [1,2]**

[1]   Department of Medicine, University of Toronto, Toronto, ON M5G 1V7, Canada; anca.prica@uhn.ca (A.P.)
[2]   Division of Medical Oncology and Hematology, Princess Margaret Cancer Center,
     Toronto, ON M5G 1V7, Canada
[*]   Correspondence: hayeong.rho@mail.utoronto.ca

**Abstract:** The standard treatment for Diffuse Large B-Cell Lymphoma (DLBCL) is rituximab, cyclophosphamide, doxorubicin, vincristine, and prednisone (RCHOP). However, many patients require subsequent treatment after relapsed disease. The ABC subtype of DLBCL (ABC-DLBCL) has a worse prognosis, and the PHOENIX trial explored adding ibrutinib to RCHOP for this patient population. The trial showed favorable outcomes for younger patients, and our study aimed to inform clinical decision-making via a cost-effectiveness model to compare RCHOP with and without ibrutinib (I-RCHOP). A Markov decision analysis model was designed to compare the treatments for patients younger than 60 years with ABC-DLBCL. The model considered treatment pathways, adverse events, relapses, and death, incorporating data on salvage treatments and novel therapies. The results indicated that I-RCHOP was more cost-effective, with greater quality-adjusted life years (QALY, 15.48 years vs. 14.25 years) and an incremental cost-effectiveness ratio (ICER) of CAD 34,111.45/QALY compared to RCHOP only. Sensitivity analyses confirmed the model's robustness. Considering the high market price for ibrutinib, I-RCHOP may be more costly. However, it is suggested as the preferred cost-effective strategy for younger patients due to its benefits in adverse events, overall survival, and quality of life. The decision analytic model provided relevant and robust results to inform clinical decision-making.

**Keywords:** diffuse large B-cell lymphoma; ibrutinib; RCHOP; Markov; cost-effective analysis; PHOENIX

## 1. Introduction

Diffuse Large B-Cell Lymphoma (DLBCL) is the most common type of Non-Hodgkin's Lymphoma (NHL), accounting for 30 to 40% of all NHL cases worldwide [1–6]. The therapeutic trajectory for DLBCL underwent a transformative shift over two decades ago with the introduction of the RCHOP regimen. This protocol includes rituximab, cyclophosphamide, doxorubicin, vincristine, and prednisone. This regimen became the first-line chemoimmunotherapy therapy for DLBCL and achieved a drastic enhancement in progression-free survival (PFS) and overall survival (OS) [7–9].

Unfortunately, a sizeable cohort with DLBCL, ranging from 30 to 40% of patients, will still experience relapsed and/or refractory disease, necessitating subsequent therapeutic interventions. Furthermore, the overall long-term survival of DLBCL patients is still only 60% [10]. Consequently, there is a persistent demand for and ongoing pursuit of more effective first-line treatments, particularly tailored to high-risk patients who are more susceptible to relapsed and/or refractory disease.

The disease spectrum of DLBCL is stratified based on varying stages of B-cell differentiation, resulting in numerous molecular subtypes [11,12]. These include the germinal center B-cell-like (GCB), activated B-cell-like (ABC), and unclassified subtypes, determined via gene expression profiling (GEP)-based methods [13]. It is notable that GEP is not commonly

used in clinical practice and that immunohistochemical methods such as the Hans algorithm are more broadly utilized for the binary distinction of GCB or non-germinal center B-cell-like (non-GCB), which includes both ABC and unclassified subtypes by GEP [14]. The molecular subtypes have prognostic implications, with the ABC subtype being associated with inferior clinical outcomes compared to the GCB subtype DLBCL [15].

Studies aimed at enhancing the outcomes and survival in patients with high-risk DLBCL have been ongoing. These endeavors have particularly included efforts to augment conventional front-line immunochemotherapy with RCHOP by incorporating innovative therapeutic agents.

In the pursuit of optimizing strategies for DLBCL, ibrutinib has emerged as a particular agent of interest. It is designed as a selective first-generation Bruton's tyrosine kinase inhibitor (BTKi) and was initially developed in 2008, subsequently gaining market availability. The BTK inhibitor was later approved for mantle cell lymphoma, served as a first-line treatment for chronic lymphocytic leukemia, and was applied in Waldenström's macroglobulinemia [16–18]. The associated side effects and toxicity profile of ibrutinib encompasses bleeding, arthralgia, hypertension, atrial fibrillation, ventricular arrhythmias, and, rarely, sudden death [16,19–21]. The increased incidence of bleeding encompasses subdural hematomas, gastrointestinal bleeding, and hematuria, although the majority of cases are characterized as grade I to II bleeding, featuring petechiae and contusions [19,20]. Atrial fibrillation is the most common cause of ibrutinib discontinuation or dose reduction to use anticoagulation [22–24]. Ventricular arrhythmia, although rare, has been reported, with a higher incidence of sudden cardiac death of 788 events per 100,000 person-years for patients, compared to 200–400 events in the age-matched population [21,25].

Ibrutinib plays a critical role in disrupting the BCR signaling pathway, which is central to the pathogenesis of the ABC subtype of DLBCL [26,27]. This subtype utilizes chronically active BCR signaling, sustained by nuclear factor kappa B (NF-κB), which is a protein transcription factor that promotes cell proliferation [28]. Recurrent mutations preventing NF-κB regulation lead to the constitutive activation of the factor. This is considered a hallmark feature of ABC-DLBCL in lymphomagenesis [29–31]. Ibrutinib's mode of action intervenes to reduce the NF-κB pathway activity, consequently inhibiting chronically active BCR signaling. This leads to the downstream effect of eliminating ABC-DLBCL cells [27].

Considering the pathophysiology of ABC-DLBCL, characterized by selective mutations to facilitate the chronically active BCR pathway, it was hypothesized that higher-risk patients with the ABC subtype may benefit from ibrutinib [27,32]. A phase I/II study was designed to evaluate single-agent ibrutinib in relapsed and/or refractory DLBCL and demonstrated preferential activity in ABC-DLBCL, with an overall response rate (ORR) of 37% [33].

A randomized controlled trial (RCT) published in 2019, named the PHOENIX trial, enrolled a cohort of 838 patients. The trial assessed and compared RCHOP with ibrutinib (n = 419) to RCHOP alone (n = 419) for the treatment of ABC-DLBCL [34]. The study showed that the addition of ibrutinib did not improve the event-free survival (EFS) for all patients. However, the subgroup analysis yielded favorable outcomes with the incorporation of ibrutinib for patients younger than 60 years (n = 342), as compared to worse outcomes for patients older than 60 years (n = 496), particularly attributed to heightened toxicity.

The current data suggests that ibrutinib may provide survival benefits in specific patient populations, and our study aimed to contribute insights to clinical decision-making. To achieve this, we employed a cost-effectiveness analysis model, comparing the therapeutic regimens and associated costs of RCHOP with and without the addition of ibrutinib. The particular focus of the investigation included patients diagnosed with ABC-DLBCL who are 60 years of age or younger. The process aimed to incorporate the effect of each approach on quality of life to highlight patient experiences throughout the treatment journey. By integrating economic and patient-centered perspectives, this study aims to provide a framework that empowers clinicians to make informed decisions based on evidence.

## 2. Methods

### 2.1. Patient Populations and Interventions

A Markov decision analytic model was designed to compare two strategies: (1) ibrutinib as a front-line intervention in adjunction to RCHOP (will be referred to as I-RCHOP) and (2) RCHOP alone (will be referred to as RCHOP), in a hypothetical cohort of patients 60 years of age or younger, who are newly diagnosed with ABC-DLBCL.

### 2.2. Model Design

The hypothetical patients underwent either I-RCHOP or RCHOP alone treatments and experienced adverse events (e.g., febrile neutropenia, nausea, and vomiting), progression-free survival, relapse with possible subsequent treatments, or death. The patients experiencing relapse were assumed to be eligible for autologous stem cell transplantation given their young age and were transitioned to salvage therapy with gemcitabine, dexamethasone, and cisplatin (GDP); those who responded to salvage therapy proceeded to autologous stem cell transplantation (auto SCT), as this is still the best available second-line therapy in Canada, while those who did not respond proceeded to a chimeric antigen receptor T-cell (CAR-T) therapy or palliative care for the last six months of their lives.

The model was evaluated on a three-month cycle per transition from one health state to another, measured in probabilities. At the end of each cycle, the hypothetical patients were susceptible to death from the disease itself, treatment complications, or natural causes at a rate derived from Statistics Canada life tables [35]. It was assumed that, after 5 years of disease remission, the risk of relapse plateaued, per published data and clinical expertise. The model was evaluated for a lifetime horizon of 30 years.

A simplified Markov decision-making model is available in Supplementary Figure S1a,b. This was aimed to simulate the clinical course of hypothetical patients 60 years of age or younger.

### 2.3. Transition Probabilities

Data for baseline probabilities of salvage treatment, autologous stem cell transplantation, and novel therapies for relapsed and/or refractory disease were derived from RCTs, including the PHOENIX trial [34]. Similarly, the probabilities of transitioning between states were extracted from published trials on standard-risk patients (Table 1). Data were obtained from survival curves using R Studio (Version 4.6) software (Posit, Boston MA, USA) [36].

**Table 1.** Baseline probabilities by variable.

| Probability | Source | Estimate | Lamba (λ) | Gamma (k) | Distribution |
|---|---|---|---|---|---|
| Probability of overall survival (I-RCHOP) | Wilson et al., 2021 [37] | - | 0.016 | 0.493 | Weibull |
| Probability of adverse events (I-RCHOP) | Younes et al., 2019 [34] | 0.642 | | | |
| Probability of event-free survival (I-RCHOP) | Wilson et al., 2021 [37] | - | 0.181 | 1.350 | Weibull |
| Probability of overall survival (RCHOP) | Wilson et al., 2021 [37] | - | 0.081 | 0.769 | Weibull |
| Probability of adverse events (RCHOP) | Wilson et al., 2021 [37] | 0.303 | - | - | - |
| Probability of event-free survival (RCHOP) | Wilson et al., 2021 [37] | - | 0.187 | 0.884 | Weibull |
| Probability of disease-free state to persistent disease (RCHOP) | Crump et al., 2017 [38] | 0.200–0.500 | - | - | - |
| Probability of overall survival from GDP | Crump et al., 2014 [39] | | 0.0638 | 0.708 | Weibull |

**Table 1.** *Cont.*

| Probability | Source | Estimate | Lamba (λ) | Gamma (k) | Distribution |
|---|---|---|---|---|---|
| Probability of salvage treatment to response | Crump et al., 2014 [39] & Crump et al., 2017 [38] | 0.300–0.451 | - | - | - |
| Probability of progression-free survival from GDP | Crump et al., 2014 [39] | - | 0.152 | 0.603 | Weibull |
| Probability of transplantation to CAR-T | Di Blasi et al., 2021 [40] | 0.201 | | | |
| Probability of disease-free transplantation to disease-free state | Crump et al. Crump et al., 2017 [38] | 0.419 | - | - | - |
| Probability of disease-free transplantation to relapse | Crump et al. Crump et al., 2017 [38] | 0.500 | - | - | - |
| Probability of disease-free transplantation to relapse to palliation | Crump et al. Crump et al., 2017 [38] | 0.393 | - | - | - |
| Probability of CAR-T to survival | Sermer et al., 2020 [41] | - | 0.056 | 0.880 | Weibull |
| Probability of CAR-T to persistent disease | Sermer et al., 2020 [41] | - | 0.169 | 0.658 | Weibull |
| Probability of CAR-T disease-free state to death | Tomas et al., 2021 [42] | 0.059 | - | - | - |
| Probability of CAR-T disease-free state to relapse | Di Blasi et al., 2021 [40] | - | 0.150 | 0.876 | Weibull |

Abbreviations: CAR-T, chimeric antigen receptor T-cell; I-RCHOP, ibrutinib in addition to rituximab, cyclophosphamide, doxorubicin, vincristine, and prednisone; GDP, gemcitabine, dexamethasone, cisplatin; RCHOP, rituximab, cyclophosphamide, doxorubicin, vincristine, and prednisone.

## 2.4. Costs and Utilities

Costs were obtained from a Canadian public health payer's perspective (CAD 1 = USD 0.78), and inflation adjusted for 2022 (Supplementary Table S1). Cancer Care Ontario unit costs served as a resource for drug acquisition costs, including standard and salvage chemotherapy, CAR-T, and supportive drugs. Human resources such as pharmacy and nursing costs were acquired from hospital administration sources. The costs of clinical visits involving physician(s) and laboratory and radiological testing were derived from the Ontario Schedule of Benefits for Physician Services, updated for 2022 fees [43]. The estimated costs for adverse events such as febrile neutropenia and nausea or vomiting, transplantation, and palliative care were obtained from a literature review. A literature review and expert opinion of health state utilities were performed. Utility value assigned to each health state (scale of 0 [death] to 1 [perfect health]), which changes with time, helps to inform the patient's journey in quality-adjusted life years (QALYs). The acquired values were incorporated into the model (Supplementary Table S2).

## 2.5. Sensitivity Analyses and Assumptions

The patient populations were selected based on available RCTs and, due to their younger age, were assumed to be transplant and CAR-T-eligible. Triage Pro R2 Release 2.1 (TreeAge Software, Williamstown, MA, USA) was utilized to create the Markov decision-making model. An annual discount rate of 1.5% to both QALYs and costs was applied as per Canadian Agency for Drugs and Technologies in Health (CADTH) guidelines [44]. A life cycle was applied. Distributions were used to perform probabilistic analyses in Table 2. One-way deterministic sensitivity analyses were conducted to show the variability of the model and included variables such as probabilities of disease state (disease-free, relapse or death) and treatment adverse events, costs of each therapy, toxicity, and palliation; and utilities of peri- and post-therapies including RCHOP, I-RCHOP, salvage chemotherapy, autologous stem cell transplant, CAR-T, and associated adverse events.

**Table 2.** Outcomes of 30-year probabilistic model with 10,000 simulations.

| Strategy | LYs, Years | SD LYs | QALYs, Years (95% CI) | SD QALYs | Costs, USD (95% CI) | ICER, USD Gained per QALY |
|---|---|---|---|---|---|---|
| RCHOP only | 17.59 | 0.01 | 14.245 (14.244–14.247) | 0.07 | 32,520.82 (32,483.85–32,557.80) | |
| I-RCHOP | 20.13 | 0.01 | 15.479 (15.478–15.480) | 0.08 | 74,606.41 (74,534.45–74,678.37) | 34,111.45 |

Abbreviations: CI, confidence interval; ICER, incremental cost-effectiveness ratio; I-RCHOP, ibrutinib in addition to rituximab, cyclophosphamide, doxorubicin, vincristine, and prednisone; LY, life years; RCHOP, rituximab, cyclophosphamide, doxorubicin, vincristine, and prednisone; SD, standard deviation; QALY, quality-adjusted life years.

## 3. Results

Probabilistic analyses of 10,000 simulations demonstrated that, for a willingness to pay (WTP) threshold of CAD 100,000/QALY, based on the 30-year horizon, the addition of ibrutinib to RCHOP chemotherapy in younger patients with ABC-DLBCL was more cost-effective (Figure 1) [45].

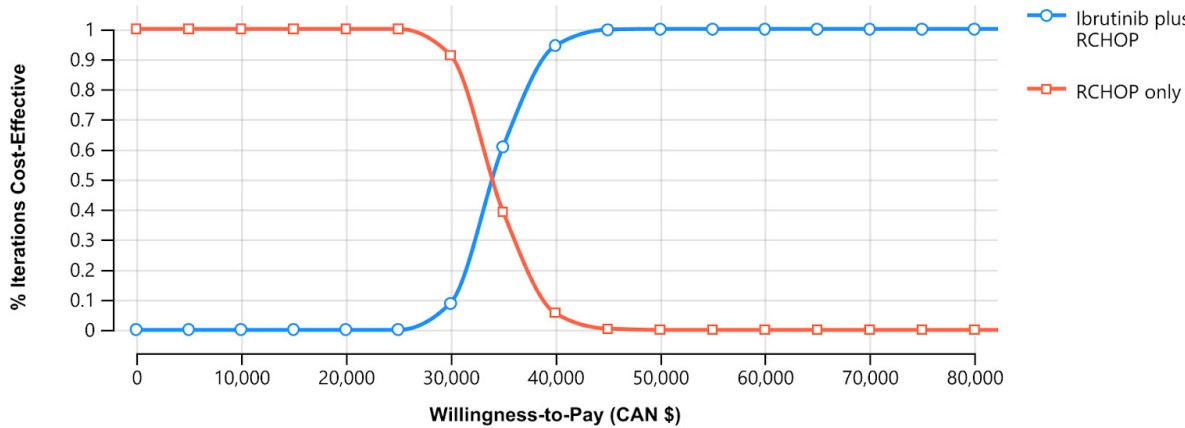

**Figure 1.** Cost-effectiveness acceptability curve between standard chemoimmunotherapy rituximab, cyclophosphamide, doxorubicin, vincristine, prednisone, and ibrutinib in addition to RCHOP (I-RCHOP). Notes: the blue line indicates the RCHOP regimen, and the red line indicates the I-RCHOP regimen. Abbreviations: CAN, Canadian; RCHOP, rituximab, cyclophosphamide, doxorubicin, vincristine, and prednisone.

The experimental strategy, I-RCHOP, was associated with greater LYs (20.13 years [SD: 0.01] vs. 17.59 years [SD: 0.01]), QALYs (15.48 years [SD: 0.08] vs. 14.25 years [SD: 0.07]). The incremental cost-effectiveness ratio (ICER; cost gained per QALY) was CAN USD 34,111.45/QALY when compared with the RCHOP-only strategy (Table 2).

Model variability and uncertainty were assessed through probabilistic sensitivity analyses over 10,000 simulations. Sensitivity analyses included key variables, such as the probability of adverse events, mortality associated with relapse, and next-line treatment strategies (Table 2).

The model was interrogated and found to be robust to increasing uncertainty surrounding probabilities by 10%. This is shown via the tornado diagram, which demonstrates a one-way sensitivity analysis comparing I-RCHOP versus RCHOP treatment regimens (Figure 2). The WTP ratio of CAD 100,000 was not crossed regardless of the values chosen for the depicted variables. The I-RCHOP strategy was robust to sensitivity analysis and, therefore, remained the most cost-effective strategy.

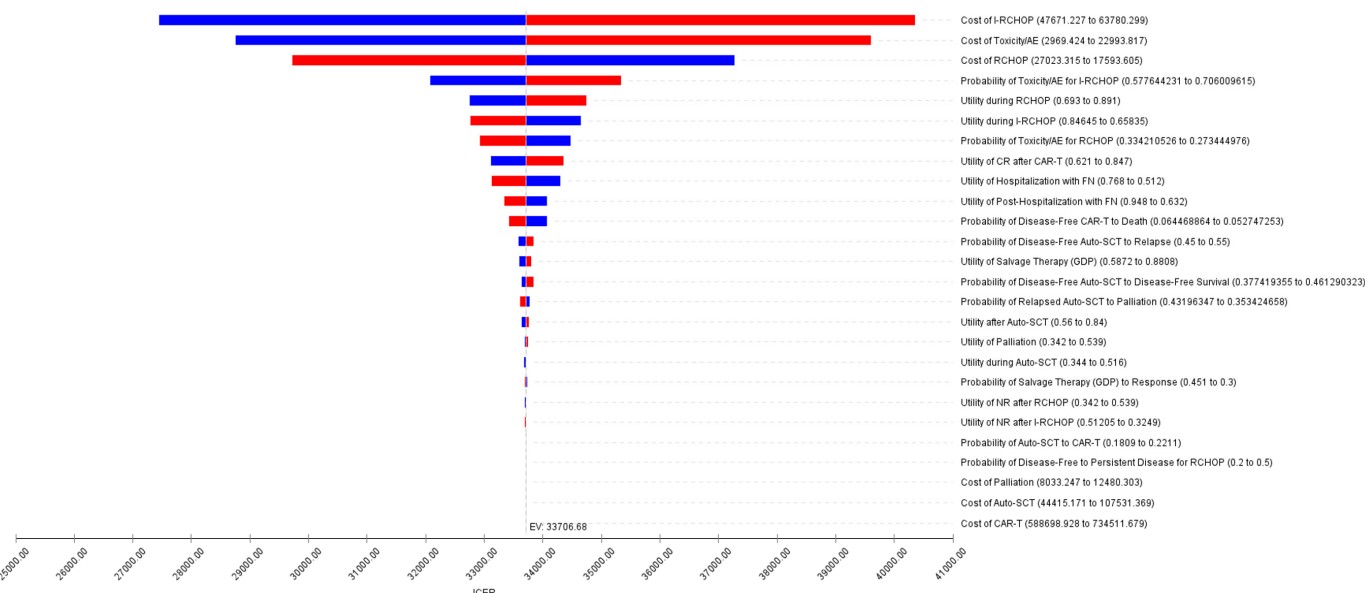

**Figure 2.** One-way sensitivity analyses show the variability of the model (tornado). The outcome is most sensitive to variation in the cost of I-RCHOP and least sensitive to variation in the cost of CAR-T. ICER is measured in Canadian dollar amount gained per quality-adjusted life years (QALY). Abbreviations: AE, adverse events; CAR-T, chimeric antigen receptor T-cell; CR, complete response; EV, expected value; FN, febrile neutropenia, GDP, gemcitabine, dexamethasone, cisplatin; ICER, incremental cost-effectiveness ratio; I-RCHOP, ibrutinib in addition to rituximab, cyclophosphamide, doxorubicin, vincristine, and prednisone; NR, non-response; SCT, stem cell transplant. Upper bound values are indicated in red, and lower bound values are indicated in blue.

## 4. Discussion

A growing body of evidence highlights the recognized limitations of current first-line standard chemoimmunotherapy RCHOP to treat DLBCL in achieving universal effectiveness [46]. This acknowledgment is particularly pronounced when comparing the outcomes across variable patient populations. Specifically, the patient population of the ABC subtype, in contrast to the GBC subtype, has shown disparities in outcomes, displaying a pattern of poorer overall survival (OS) and progression-free survival (PFS) [12,15].

This observed disparity in clinical responses among DLBCL subtypes highlights an intrinsic heterogeneity within the disease and emphasizes the need for a refined and precisely targeted therapeutic approach. This distinction of subtypes is made possible with the utilization of various investigations, inclusive of gene expression profiling (GEP), as used in the PHOENIX trial, or immunohistochemistry-based methods, such as the Hans algorithm, which is more commonly used in clinical practice [14,34]. As the landscape of therapeutic innovation continues to evolve with the development of novel agents for DLBCL, the optimization of targeted therapy tailored to specific DLBCL subtypes is becoming more important in practitioner and patient decision-making processes.

In consideration of refined subtypes of DLBCL and their diverse outcomes, an addition of BTKi was suggested to target the distinct pathological pathway associated with the disease [32]. A subgroup analysis of the PHOENIX trial revealed that individuals under 60 years of age and with the ABC subtype derived notable benefits from the integration of ibrutinib with RCHOP [34]. The data from the Phase III trial motivated our cost-effectiveness analysis, which aimed to assess the health and economic viability of adding ibrutinib to the traditional RCHOP chemoimmunotherapy regimen.

At a WTP of CAD 100,000/QALY, the I-RCHOP strategy was a cost-effective treatment when compared with RCHOP. This approach was demonstrated via a Markov decision-making model to be dominant over a 30-year horizon when considering higher life year numbers, as well as quality-adjusted years measured in QALYs. This analysis was per-

formed by incorporating the high costs of salvage therapies such as stem cell transplantation and CAR-T therapy in Canadian healthcare settings. Multiple sensitivity analyses indicated that our model was robust.

I-RCHOP would be a more costly treatment regimen given the currently high market price for ibrutinib in Canada; however, it was assessed to be the overall preferred cost-effective treatment strategy for younger patients with ABC-DLBCL. Given the increasing availability of generic medications, we anticipate that I-RCHOP, with its promising treatment profile in this subset of the patient population will become more cost-effective. It is also notable that ibrutinib is an available drug in some jurisdictions across the world and could have an accessible price point, particularly as it approaches becoming a generic drug. Although the formulation of ibrutinib (Imbruvia®, Pharmacyclics LLC, South San Francisco, CA, USA) has a higher price point in high-income country markets, exemplified by its cost in Canada, low-income countries such as India have achieved success in producing generic formulations at a lower cost [47,48]. However, it is worth noting that the cost remains a barrier for the patient population in India, as the generic formulations are still priced at twice the gross national income per capita [47].

In countries where access to CAR-T therapy for relapsed disease is confined to a restricted number of qualifying care institutions, the prospect of enhancing disease control in the front-line setting becomes more appealing from clinical practice and funding perspectives [49]. In such scenarios, there is a compelling rationale for investing efforts and resources in optimizing therapeutic strategies, and the upfront addition of ibrutinib with RCHOP may mitigate the demand for subsequent CAR-T therapy.

### 4.1. Alternative Novel Therapies

The search for alternative therapies, in addition to the standard chemoimmunotherapy, has been a persistent research endeavor in DLBCL. In this context, acalabrutinib, a second-generation BTK inhibitor, is currently being studied in the Phase III clinical trial, ESCALADE (NCT04529772) [50]. Similar to the PHOENIX trial, patients with non-GCB DLBCL and younger than 65 years old are recruited for comparison of the addition of acalabrutinib to the standard regimen to RCHOP only. With enhanced kinase selectivity, acalabrutinib has the potential for better efficacy than ibrutinib. It has also been noted to have a more favorable safety profile than ibrutinib, with lower adverse event burden overall and for atrial fibrillation, hypertension, and hemorrhage compared with ibrutinib [51,52]. As we accrue further data with increased market availability of acalabrutinib, performing a cost-effective analysis based on this trial may inform clinical decision-making, especially regarding the subgroup of patients who traditionally exhibit higher rates of relapsed and/or refractory disease.

Notably, the POLARIX (NCT03274492), which is a recently published Phase III trial, compared the modified regimen of polatuzumab vedotin, rituximab, cyclophosphamide, doxorubicin, and prednisone (pola-R-CHP) to RCHOP in patients with DLBCL [53]. This regimen substituted vincristine for polatuzumab vedotin, which is a monoclonal antibody-drug conjugate targeting CD79b. As CD79b is ubiquitously expressed on the surface of malignant B cells, the conjugate has previously been studied in relapsed and/or refractory DLBCL, and has shown safety and efficacy [54,55]. The POLARIX trial yielded results indicating that pola-R-CHP significantly reduced the risk of progression, relapse, or death in comparison to the standard RCHOP regimen (stratified hazard ratio [HR], 0.73; 95% confidence interval [CI], 0.57 to 0.95; $p = 0.002$).

With available data from the POLARIX, a cost-effectiveness analysis was performed [49]. This trial demonstrated that pola-R-CHP was more cost-effective at a WTP of USD 150,000/QALY. The study showed that the cost-effectiveness of the modified regimen was highly sensitive to the PFS, particularly in certain clinical scenarios involving refractory diseases, such as when CAR-T therapy is used as second-line therapy. It is important to acknowledge that while polatuzumab vedotin exhibits promising therapeutic potential, it stands as a

relatively recent and expensive medication. Consequently, options such as oral ibrutinib currently have broader applications worldwide.

It is worth noting that, in comparison to the PHEONIX trial, which focused on the ABC-DLBCL subgroup population, the POLARIX trial implemented a distinct treatment strategy for DLBC. The latter featured broader inclusion criteria for the study cohort without explicitly differentiating the subtypes of DLBCL [53]. The inherent variations in the design and demographics of these two trials prompt caution in undertaking a direct comparison of cost-effective analyses. However, as we observe a rise in promising and positive outcomes associated with novel agents, future cost-effectiveness studies may consider an indirect comparison that incorporates these innovative therapeutics into RCHOP/CHP regimens. This approach may yield a comprehensive understanding of evolving DLBCL treatment options and their economic implications, ultimately guiding informed decision-making in clinical practice.

### 4.2. Significance of Precision Medicine

A recent sub-analysis of the PHOENIX trial demonstrated the potential utility of incorporating RNA sequencing for assessing *MYC* and *BCL2* co-expression [56]. This serves as an adjunct test to the immunohistochemistry-based Hans algorithm and/or gene expression profiling. This integrated approach further refines the initial categorization of patients into GCB and ABC subtypes and allows for the assessment of their clinical outcomes, specifically based on the genetic rearrangements that characterize high-grade B-cell lymphomas [57,58]. The event-free survival (EFS) exhibited superiority in the population with high *MYC/BCL2* co-expression, with no discernible impact on the overall survival (OS) of the group. Analyzing this cohort based on age, the study demonstrated that patients under the age of 60 years (n = 239) significantly benefited from the addition of ibrutinib (EFS: hazard ratio [HR], 0.58; 95% confidence interval [CI], 0.38–0.88; $p$ = 0.0099; OS: HR, 0.33; 95% CI, 0.16–0.67; $p$ = 0.0013). The findings of this study suggest that implementing RNA sequencing may offer potential benefits in tailoring the treatment approach for the specific age group of patients. However, it is important to acknowledge that there is uncertainty in determining the cost of RNA sequencing associated with each care institution, which may limit the widespread adaptation of this testing method; this may influence the overall decision-making process. This consideration is further reinforced by a study that performed a cost-effective analysis of a precision medicine treatment strategy for DLBCL [59]. In this analysis, the improved accuracy provided by more precise methods, such as gene expression profiling, did not result in a significant compromise of health outcomes. The constrained accessibility of high-cost investigations did not outweigh the benefit achieved through the incorporation of novel agents, contributing to improved survival outcomes across all patient groups. This finding may be relevant to the Canadian healthcare system, where resources allocated to the utilization of precision medicine for both diagnosis and treatment are frequently confined to a limited number of expert care centers [60].

### 4.3. Strengths and Limitations

There are several limitations within our analysis, including the limited number of clinical trials used to construct our model. Our analysis relied heavily on the efficacy results of the PHOENIX trial. As this trial was completed in 2020, the long-term follow-up data pertaining to disease relapse and subsequent steps in treatment trajectory, such as salvage chemotherapy, autologous stem cell transplant, CAR-T therapy, or palliative care, have not been published for the patient population younger than 60 years of age who had completed I-RCHOP. To address these data gaps, probability values from other trials encompassing a broader, non-age limited, non-subtyped population were incorporated into the model. At the time of model development, CAR-T therapy was still considered a novel therapeutic, and therefore, there is limited pertinent published data. Future studies may consider incorporating CAR-T therapy data as a second-line approach, particularly if it evolves into a standardized, funded treatment option in the Canadian healthcare landscape.

The decision to set a 30-year horizon for our model necessitated data extrapolation beyond the 5-year data available from clinical trials. This specific threshold was chosen as our patient population was younger than 60 years old, with a longer life span derived from a baseline representation of the general Canadian population [35]. Additionally, the availability of extended follow-up data on the prognosis of patients in sustained remission for 5 years or more allows for reliable modeling of decision analysis [9,38,61,62].

Cost and utility values were obtained from both existing literature and expert opinions. Although recognizing the potential for bias in treatment decision-making pertaining to a subset of patients experiencing relapsed and/or refractory disease, we note that the bias likely is minimized due to a relatively smaller proportion of patients transitioning beyond the disease-free state in our analysis. The overall conclusion derived from this model remained unaffected. This stability can be attributed to the consistent trajectory of both treatment arms followed, converging into salvage chemotherapy and autologous stem cell transplant once the disease is confirmed to be relapsed and/or refractory.

Our cost-effective analysis demonstrates notable strengths arising from a comprehensive search process and incorporation of data from pertinent randomized controlled trials, particularly those aligned with the PHOENIX trial. The inclusion of real-world data, reflective of survival outcomes and costs within the Canadian healthcare system, further enhances the credibility of the study. Additionally, the findings suggest the feasibility of conducting economic evaluations utilizing existing provincial oncologic databases. This study advocates for the exploration of alternative funding approaches, offering insights for decision-makers in negotiating drug prices and fostering the long-term sustainability of healthcare systems.

## 5. Conclusions

In conclusion, our cost-effective analysis indicates that the addition of ibrutinib to the standard RCHOP chemoimmunotherapy regimen was more cost-effective than RCHOP alone, optimizing overall survival and QALYs. This model serves to demonstrate the intricate, practical decision-making processes undertaken by clinicians and their patients in real-life scenarios. Additionally, it may contribute to advocating for broader accessibility of chemoimmunotherapy on a systemic level, utilizing this evidence-based approach tailored for specific patient populations. Through the incorporation of key variables such as probabilities of adverse events, utilities, and associated costs of advanced treatments, our decision analytic model, while not absolute, offers pertinent and robust results to help inform clinical decision-making processes.

**Supplementary Materials:** The following are available online at https://www.mdpi.com/article/10.3390/curroncol30120764/s1, Figure S1: (**a**) Simplified Markov Model (I-RCHOP Arm), (**b**) Simplified Markov Model (RCHOP Arm); Table S1: Cost estimates used in analysis [37,63–72], Table S2: Health Utility Data [73–82].

**Author Contributions:** Conceptualization, H.R. and A.P.; methodology, H.R., and I.J.-H.J.; software, H.R., and I.J.-H.J.; validation, H.R., I.J.-H.J. and A.P.; formal analysis, H.R., and I.J.-H.J.; writing—original draft preparation, H.R., and I.J.-H.J.; writing—review and editing, H.R., I.J.-H.J. and A.P.; visualization, H.R.; supervision, A.P.; project administration, H.R.; funding acquisition, H.R., and A.P. All authors have read and agreed to the published version of the manuscript.

**Funding:** This research was funded by the American Society of Hematology (ASH) through Hematology Opportunities for the Next Generation of Research Scientists Award (HONORS Award).

**Institutional Review Board Statement:** Ethical review and approval were waived for this study due to the data presented in this study are publicly available.

**Informed Consent Statement:** Not applicable.

**Data Availability Statement:** The data presented in this study are openly available: https://www.ncbi.nlm.nih.gov/pmc/articles/PMC6553835/ (accessed on 14 October 2022).

**Acknowledgments:** We would like to acknowledge Pamela Ng and Felicia Leung for their assistance in obtaining cost information for chemotherapy. We appreciate Abi Vijenthira for their advice on cost-effectiveness analysis.

**Conflicts of Interest:** The authors declare no conflict of interest.

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
