# Peer review of "Ibrutinib Plus RCHOP versus RCHOP Only in Young Patients with Activated B-Cell-like Diffuse Large B-Cell Lymphoma (ABC-DLBCL): A Cost-Effectiveness Analysis"

_curroncol, doi:10.3390/curroncol30120764_

Round 1
Reviewer 1 Report
Comments and Suggestions for Authors
In the present study, Rho et al. showed a comparative analysis of I-RCHOP vs RCHOP treatment outcomes among young ABC-DLBCL patients. The study is interesting and of potential interest in the field and showed advancement in the treatment of ABC-DLBCL. I have a few minor questions which need to be addressed. The comments are as follows:
1. What are the side effects of Ibrutinib in the patients? The authors should elaborately discuss.
2. How feasible is the use of I-RCHOP in low to middle income countries?
3. Figure 2: Axis labels and writing should be enlarged as it is not at all visible.
4. The study is based on Canadian cohort. How different the data could be when compared to the other parts of the world?
5. What could be the alternative improved treatment measures for the middle-aged and fairly elder patients?
6. Why the I-RCHOP treatment is limited to young patients only? Kindly discuss.
Author Response
Dear Reviewer of Current Oncology:
Thank you for considering our article, “Ibrutinib plus RCHOP versus RCHOP only, in young patients with activated B cell-like diffuse large B-cell lymphoma (ABC-DLBCL): a cost effectiveness analysis”, for publication on Current Oncology.
The following is the point-to-point response to your revisions:
In the present study, Rho et al. showed a comparative analysis of I-RCHOP vs RCHOP treatment outcomes among young ABC-DLBCL patients. The study is interesting and of potential interest in the field and showed advancement in the treatment of ABC-DLBCL. I have a few minor questions which need to be addressed. The comments are as follows:
- What are the side effects of Ibrutinib in the patients? The authors should elaborately discuss.
- We thank you for your comment. This has been completed. We have added the line: [Ibrutinib’s] side effects and toxicity profile includes bleeding, arthralgia, hypertension, atrial fibrillation, ventricular arrhythmias and rarely, sudden death.
- We have revised the manuscript accordingly.(line 61-69)
- How feasible is the use of I-RCHOP in low to middle income countries?
- We thank you for your comment. While the ibrutinib (IMBRUVICA®) formulation by pharmaceutical company, Janssen, still remains expensive in the high-income country markets (reflected cost in Canada, specifically), low income countries such as India have been successful in developing generic formulations for lower cost.1,2 The cost, however, is still prohibitive for the general population as the generic formulations are still twice the gross national income of the population per capita.1
- We have revised the manuscript accordingly. (line 230-238)
- Figure 2: Axis labels and writing should be enlarged as it is not at all visible.
- We thank you for your comment. This has been completed. We had submitted a separate file for higher quality JPEG of the diagram.
- The study is based on Canadian cohort. How different the data could be when compared to the other parts of the world?
- We thank you for your comment. From our literature search, I-RCHOP regimen for cost-effectiveness has only been studied in Canada. As the costs and drug availability of this chemoimmunotherapy regimen are different depending on funding delivery models of each country, we hypothesize that the data will reflect this variation.
- What could be the alternative improved treatment measures for the middle-aged and fairly elderly patients?
- We thank you for your comment. With Canadian and international guidelines, RCHOP remains the standard of care regimen for patient populations across variable demographics. Dose-reduced RCHOP (R-miniCHOP) can be considered for very elderly or frail patients, depending on their disease risks and comorbidities.
- We also have noted that acalabrutinib, a second-generation BTK inhibitor with reduced toxicities for elderly patients, is currently undergoing a Phase III randomized controlled trial. The outcomes are highly anticipated upon the study's completion, and although the data are expected to be published in a few years, we eagerly await their release. (line 248-259)
- Why the I-RCHOP treatment is limited to young patients only? Kindly discuss.
- We thank you for your comment. The PHOENIX trial (2019) has demonstrated that younger patients with ABL DLBCL less than 60 years of age had improvement in their survival, when ibrutinib was added to the standard regimen of RCHOP.
- In elderly, frail patients, the combination of ibrutinib and standard chemotherapy was not as well-tolerated in terms of side effects and toxicity compared to the younger population. This observation reinforces the notion that adding ibrutinib to RCHOP may be more suitable for younger patients.
Thank you again for considering our manuscript.
Sincerely,
Hayeong Rho
Irene Jeong
Anca Prica (Principal Investigator)
References
(1) Singh, C.; Jindal, N.; Youron, P.; Malhotra, P.; Prakash, G.; Khadwal, A.; Jain, A.; Sreedharanunni, S.; Sachdeva, M. U. S.; Naseem, S.; Varma, N.; Varma, S.; Lad, D. P. Efficacy, Safety, and Quality of Life of Generic and Innovator Ibrutinib in Indian CLL Patients. Indian J Hematol Blood Transfus 2021, 37 (2), 313–317. https://doi.org/10.1007/s12288-020-01378-6.
(2) Hegde, N. C.; Kumar, A.; Kaundal, S.; Saha, L.; Malhotra, P.; Prinja, S.; Lad, D.; Patil, A. N. Generic Ibrutinib a Potential Cost-Effective Strategy for the First-Line Treatment of Chronic Lymphocytic Leukaemia. Ann Hematol 2023, 102 (11), 3125–3132. https://doi.org/10.1007/s00277-023-05342-y.

Reviewer 2 Report
Comments and Suggestions for Authors
This article tried to demonstrate that the addition of Ibrutinib to RHOP can improve the outcome for patients with ABC DLBCL less than 60 years. The rationale was provided by the RCT Phoenix where a subset of ABC DLBCL was improved with the addition of ibrutinib.
From these data they gather some other trials to built a model evaluating the adjunction of Ibrutinib to R CHOP in a population 60 years of age new diagnoses ABC-DLBCL.. This model demonstrated complex decision making pathways and provided robust results for adding Ibrutinib to RCHOP.!
Main comments:
the complexity of the model and the analytic tools overcome the standard understanding of clinician, who are responsible for the introduction of Ibru.
The difference on outcome for the I RCHOP is a result of a retrospective analysis on a relatively limited number of patients. It was not a prospective randomization. Same remarks for the other trials.
For clarification the total number of patients treated with Ibrutinib in each main analysis should be present: line 55 and table 1; line 192
Moreover, the prognostic value of ABC versus GCB is not found in all studies and is also reliable to the methods used ( gene profiling, vs immunochemistry….)
Cost estimation is just for Canada. Each country has its own ways of looking at the cost of expensive drugs line 97 and other parameters
Line 111 population from other RCT. Do you have individual patients data
The 30 years horizon had been set for this model line221. I am not sure that it is possible for most studies,. If yes, provide the follow up after 5 years with the number of patients still on the road.
Author Response
Dear Reviewer of Current Oncology:
Thank you for considering our article, “Ibrutinib plus RCHOP versus RCHOP only, in young patients with activated B cell-like diffuse large B-cell lymphoma (ABC-DLBCL): a cost effectiveness analysis”, for publication on Current Oncology.
The following is the point-to-point response to your revisions:
The complexity of the model and the analytic tools overcome the standard understanding of clinicians, who are responsible for the introduction of Ibru.
The difference on outcome for the I RCHOP is a result of a retrospective analysis on a relatively limited number of patients. It was not a prospective randomization. Same remarks for the other trials.
For clarification the total number of patients treated with Ibrutinib in each main analysis should be present: line 55 and table 1; line 192
- This was completed in the revised manuscript.
- For the PHOENIX trial, a total of 838 patients were enrolled, divided equally to 2 groups for each treatment arm (n=419 each); within the participant populations, there were 342 individuals younger than 60 years, and 496 older than 60 years. (line 83-89)
- For sub-analysis of the PHOENIX trial, patients with ABC DLBCL who are younger than 60 years (n=239) were specified. (line 299-302)
- For table 1, we note that the table highlights the baseline probabilities by variable. We have instead incorporated the number of patients enrolled in the study within the bodies of the paragraphs. (line 83-89, line 299-302)
Moreover, the prognostic value of ABC versus GCB is not found in all studies and is also reliable to the methods used ( gene profiling, vs immunochemistry….)
- We agree, the categorization of each patient population to ABC vs GCB is not present in all studies we have acquired data from, and the extrapolated data for those who have remained in remission (not subtyped) have been incorporated into our iterations of analysis.
Cost estimation is just for Canada. Each country has its own ways of looking at the cost of expensive drugs line 97 and other parameters
- We have noted that the cost estimation is from Canada, which may vary from other countries with variable gross national income (also per capita).
- We have revised the manuscript accordingly. (line 230-238)
Line 111 population from other RCT. Do you have individual patients data
- No, we do not have access to individual patient-level data. Due to our limited access, we relied on aggregate data from various publications.
The 30 years horizon had been set for this model line221. I am not sure that it is possible for most studies. If yes, provide the follow up after 5 years with the number of patients still on the road.
- We thank you for your comment. As we modelled treatment options beyond the trial follow-up period, considering robust extended follow-up data for ASCT, CART, and RCHOP chemotherapy from other studies, we feel it is most appropriate to simulate a 30-year horizon.1–3 This approach aims to mirror the lifelong clinical course of these younger patients in our model.
- We have revised the manuscript accordingly. (line 331-336)
Thank you again for reviewing our manuscript.
Sincerely,
Hayeong Rho
Irene Jeong
Anca Prica (Principal Investigator)
References
(1) Sermer, D.; Batlevi, C.; Palomba, M. L.; Shah, G.; Lin, R. J.; Perales, M.-A.; Scordo, M.; Dahi, P.; Pennisi, M.; Afuye, A.; Silverberg, M. L.; Ho, C.; Flynn, J.; Devlin, S.; Caron, P.; Hamilton, A.; Hamlin, P.; Horwitz, S.; Joffe, E.; Kumar, A.; Matasar, M.; Noy, A.; Owens, C.; Moskowitz, A.; Straus, D.; von Keudell, G.; Rodriguez-Rivera, I.; Falchi, L.; Zelenetz, A.; Yahalom, J.; Younes, A.; Sauter, C. Outcomes in Patients with DLBCL Treated with Commercial CAR T Cells Compared with Alternate Therapies. Blood Advances 2020, 4 (19), 4669–4678. https://doi.org/10.1182/bloodadvances.2020002118.
(2) Alarcon Tomas, A.; Fein, J. A.; Fried, S.; Fingrut, W.; Anagnostou, T.; Alperovich, A.; Shah, N.; Fraint, E.; Lin, R. J.; Scordo, M.; Afuye, A. O.; Batlevi, C. L.; Besser, M.; Dahi, P. B.; Danylesko, I.; Giralt, S.; Imber, B. S.; Jacobi, E.; Nagler, A.; Palomba, M. L.; Salles, G.; Sauter, C. S.; Shah, G. L.; Shem-Tov, N.; Shimoni, A.; Yahalom, J.; Yerushalmi, R.; Avigdor, A.; Perales, M.-A.; Shouval, R. Novel Agents May Be Preferable to Chemotherapy for Large B-Cell Lymphoma Progressing after CD19-CAR-T: A Multicenter Observational Study. Blood 2021, 138 (Supplement 1), 883. https://doi.org/10.1182/blood-2021-147568.
(3) Crump, M.; Neelapu, S. S.; Farooq, U.; Van Den Neste, E.; Kuruvilla, J.; Westin, J.; Link, B. K.; Hay, A.; Cerhan, J. R.; Zhu, L.; Boussetta, S.; Feng, L.; Maurer, M. J.; Navale, L.; Wiezorek, J.; Go, W. Y.; Gisselbrecht, C. Outcomes in Refractory Diffuse Large B-Cell Lymphoma: Results from the International SCHOLAR-1 Study. Blood 2017, 130 (16), 1800–1808. https://doi.org/10.1182/blood-2017-03-769620.
